# Couple oriented counselling improves male partner involvement in sexual and reproductive health of a couple: Evidence from the ANRS PRENAHTEST randomized trial

Cyprien Kengne-Nde[1,2,3,¤a]*, Mathurin Cyrille Tejiokem[1,4], Joanna Orne-Gliemann[5], Bernard Melingui[6], Paul Koki Ndombo[7], Ngo A. Essounga[8], Anne Cécile Bissek[9], Simon Cauchemez[3,4], Patrice T. Tchendjou[1,4,¤b]*

1 Centre Pasteur Cameroon, Epidemiology and Public Health Service, Yaounde, Cameroon, 2 Bordeaux School of Public Health, Bordeaux University, Bordeaux, France, 3 Mathematical Modeling of Infectious Diseases Unit, Pasteur Institute, Paris, France, 4 International Network of Pasteur Institutes, Paris, France, 5 INSERM U897, Bordeaux School of Public Health, Bordeaux University, Bordeaux, France, 6 Centre Pasteur Cameroon, Hematology Laboratory, Yaounde, Cameroun, 7 Centre of Mother and Child, Chantal Biya Foundation, Yaounde, Cameroon, 8 Ministry of Scientific Research and Innovation, Yaounde, Cameroon, 9 Division of Operational Research, Ministry of Public Health, Yaounde, Cameroon

¤a Current address: Research, Monitoring and Evaluation Unit, Regional Technical Group of National AIDS Control Committee, Douala, Cameroon
¤b Current address: Strategic Information and Evaluation Unit, EGPAF Yaounde, Yaounde, Cameroon
* cyprienkengne@gmail.com (CK-N); ttpatty@yahoo.com (PTT)

## Abstract

### Background

Male partner involvement (MPI) has been recognized as a priority area to be strengthened in Prevention of Mother to Child Transmission (PMTCT) of HIV. We explored the impact of Couple Oriented Counselling (COC) in MPI in sexual and reproductive health and associated factors.

### Method

From February 2009 to October 2011, pregnant women were enrolled at their first antenatal care visit (ANC-1) and followed up until 6 months after delivery in the Mother and Child Center of the Chantal Biya Foundation within the randomized prenahtest multicentric trial. The MPI index was defined using sexual and reproductive health behaviour variables by using multiple correspondence analysis followed by mixed classification. Men were considered as highly involved if they had shared their HIV test results with their partner, had discussed on HIV or condom used, had contributed financially to ANC, had accompanied their wife to ANC or had practiced safe sex. Factors associated to MPI were investigated by the logistic model with GEE estimation approach.

### Results

A total of 484 pregnant women were enrolled. The median age of the women was 27 years (IQR: 23–31) and 55.23% had a gestational age greater than 16 weeks at ANC-1. Among

**Data Availability Statement:** Data supporting the conclusions of this article cannot be made publicly

available due to ethical restrictions. These data are available from the Centre Pasteur of Cameroon, Epidemiology and Public Health Service, Yaounde, Cameroon, at P.O BOX: 1274 Yaoundé, 451, Rue 2005, Yaoundé 2 - Cameroun Phone: (237) 222 23 10 15 / 222 23 18 03 - 691 819 685 Email: cpc@pasteur-yaounde.org.

**Funding:** This study was supported by the Agence Nationale de Recherches sur le SIDA et les hépatites virales (French National Agency on AIDS Research) (grant ANRS 12127). Complementary funding was provided by the Elizabeth Glaser Pediatric AIDS Foundation (Sub-agreement 354–07). The funders had no role in study design, data collection and analysis, decision to publish, or preparation of the manuscript.

**Competing interests:** The authors have declared that no competing interests exist.

**Abbreviations:** AIDS, Acquired Immuno Deficiency Syndrome; ANC-1, First Antenatal care; ANRS, Agence Nationale de Recherches sur le SIDA et les hépatites virales; aOR, Adjusted Odds Ratio; ART, Antiretroviral treatment; ARV, Antiretroviral Therapy; CC, Classical Counselling; CI, Confidence Interval; COC, Couple-Oriented post-test HIV Counselling; GEE, Generalized Estimating Equation; HIV, Human Immuno Deficiency Virus; MCA, Multiple Correspondence Analysis; MPI, Male Partner Involvement; OR, Crude Odds-Ratio; PMTCT, Prevention of Mother to Child Transmission; SC, Standard post-test HIV Counselling.

them, HIV prevalence was 11.9% (95% CI: 9.0–15.4). The median duration of the women's relationship with their partner was 84 months (IQR: 48–120). MPI index at 6 months after delivery was significantly greater in the COC group than the classical counselling group (14.8% vs 8,82%; p = 0,043; Fig 1). The partners of the women who participated in the COC were more likely to be involved during follow up than others (aOR = 1.45; 95% CI = 1.00–2.10). Partners with no incoming activity (aOR = 2.90; 95% CI = 1.96–4.29), who did not used violence within the couple (aOR = 1.70; 95% CI = 1.07–2.68), and whose partner came early for ANC-1 (aOR = 1.37; 95% CI = 1.00–1.89) were more likely to be involved than others.

## Conclusion

MPI remains low in stable couples and COC improves partner involvement. Our findings also support the need of strengthening outreach towards "stable" couples and addressing barriers. This could go a long way to improve PMTCT outcomes in Cameroon.

## Trial registration

PRENAHTEST, NCT01494961. Registered 15 December 2011—Retrospectively registered, https://clinicaltrials.gov/ct2/show/NCT01494961.

## Introduction

HIV / AIDS infection remains a public health problem worldwide. In 2019, 1.7 million people were newly infected with HIV, including 150 000 children under the age of 15 [1]. In sub-Saharan Africa, five in six new infections among adolescents aged 15–19 years were girls [1]. Young women aged 15–24 years are twice likely to be living with HIV than men [1]. In Cameroon, 17 000 people were newly infected with HIV in 2019. Seventy-three per cent of pregnant women living with HIV accessed antiretroviral drugs to prevent transmission of the virus to their baby, preventing 3200 new HIV infections among newborns. Early infant diagnosis stood at 64% [2]. The recent Population-based HIV Impact Assessment (CAMPHIA) conducted in 2017 reported an HIV prevalence of 3.7% in the population aged 15–64 years [3]. The national prevalence was 5.70% (95% CI: 4.93–6.40) among pregnant women with a high incidence of cases among people who live in a stable relationship and who have already used the health care system at least once [4].

Many interventions to reduce new HIV infections have been described. These studied focused on HIV counselling and testing, antiretroviral therapy access, medical circumcision, promotion of safer sex and all strategies facilitating involvement of male partner in the couple's sexual and reproductive life which improves sexual and reproductive health outcomes [5, 6].

The definition of Male Partner Involvement (MPI) in the sexual and reproductive health of the couple varies from one study to another depending on the socio-cultural context. For Rutenberg et al., Male partner involvement (MPI) was defined as accompanying their partner to the hospital, or participating in a counselling session and being tested for HIV. MPI was also defined as providing financial support or supporting their wives to cope with HIV infection and to benefit from Prevention of Mother-To-Child Transmission (PMTCT) programs [7].

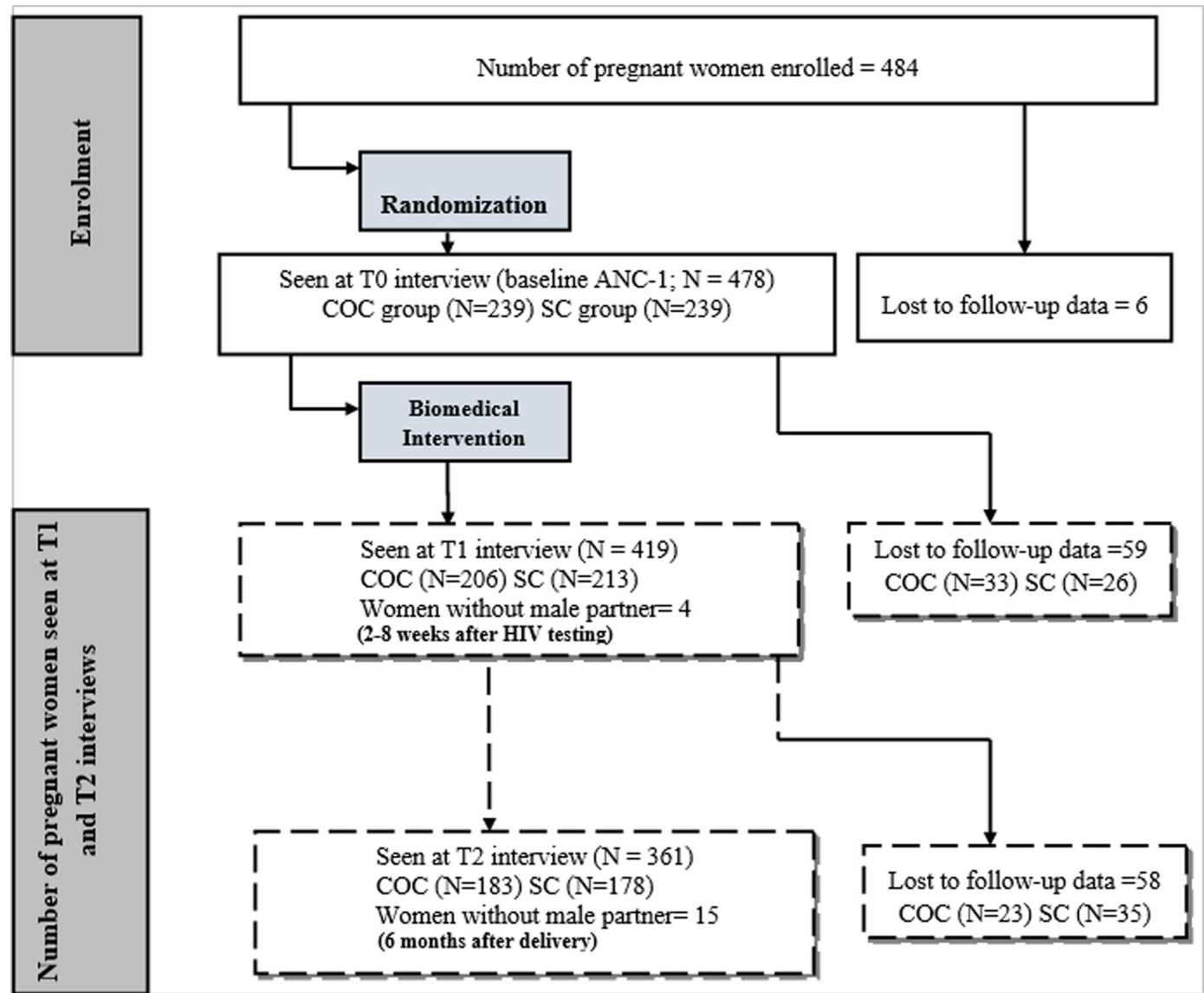

**Fig 1. Flowchart of enrolment and follow-up within the cohort of pregnant women in Cameroon, ANRS 12127-12236/Prenahtest trial (2009–2011).**

MPI has been recognized as a priority for PMTCT programs of HIV / AIDS [8]. MPI play an important role in the risk of contracting HIV virus in women [5], the use of condoms in couples in order to prevent infection [9, 10], the use of PMTCT services by women [10, 11], women's decisions regarding treatment [9, 12, 13], and the follow-up infant feeding [9, 13, 14].

In addition, when men are counselled and tested for HIV during prenatal consultations, previous studies found an increase of intake of ARV treatment or prophylaxis [13] and a decline in the MTCT rate and infant mortality [15]).

Behavioural studies suggested an improvement in the context of counselling for HIV testing and an approach favouring greater MPI. It is in this perspective that the ANRS 12127-Prenahtest project was set up in 2009. We therefore sought to explore the impact of Couple Oriented Counselling (COC) on sexual and reproductive health and analysed associated factors.

## Methods

### ANRS 12127 Prenahtest trial, study site and data collection

The Prenatal HIV Testing (Prenahtest) trial is a multi-country randomized trial aimed to assess the impact of prenatal COC on the incidence of partner HIV testing. It also aimed to assess the impact ofcouple HIV counselling, on sexual, reproductive and HIV prevention behaviours [16–19]. The Prenahtest trial was held in four countries with different sociocultural phases (Cameroon, Dominican Republic, Georgia and India). COC is a clinic-based behavioural intervention aiming to replace standard post-test HIV counselling delivered to pregnant women. This individual discussion with the pregnant woman took place during prenatal care, instead of the standard post-test counselling. The couple-oriented post-test HIV counselling session comprised of the standard post-test counselling components and couple-oriented components. It aims at providing the woman with information, building-up their negotiation skills and confidence, and giving them the tools and strategies to actively involve their partner in the prenatal HIV counselling and testing process. The end goal of COC is to facilitate the management of the HIV test results in couples.The framework of COC is based on available reference counselling module, and partly inspired by the Health Belief Model [19, 20].

In Cameroon, Prenahtest took place at Centre Mere Enfant (CME), an urban healthcare structure in care of pregnant women, located in the capital region, Yaounde. CME is a reference healthcare centre, and is accessible to woman of all social classes living in Yaounde. The services offered to mothers included gynaecological consultations, antenatal clinics with a sophisticated component of prevention of mother to child transmission of HIV.

Between February and October 2009, pregnant women who consulted for ANC-1 visit were enrolled and randomized equally into two groups of counselling (COC or CC). Interventions and data collection were conducted by trained healthcare professionals. Enrolment occurred before HIV testing. Free HIV testing using a rapid tests algorithm [Determine (1st) and Immunocomb (2nd)] was carried out on the same day of recruitment. HIV test results were given to the pregnant women during the second scheduled ANC-1 visit, planned to coincide with the appointment for other medical examinations prescribed during ANC-1. Three structured face-to-face quantitative questionnaires were administered to trial participants: at baseline prior to prenatal HIV testing (T0), 2–8 weeks after the HIV post-test counselling (T1), and 6 months post-partum (T2). The questionnaire administered at T0 documented socio-demographic characteristics, couple relationship, women violence experiences, women attitudes and practices concerning sexual behaviours before pregnancy, HIV prevention, family planning as well as history of HIV testing. At T1 and T2, the questionnaire administered documented the occurrence of their male partners' HIV testing, circumstances when it occurred, evolution of communication in their relationships, women violence experienced during or after pregnancy. Women were assigned identification numbers and all the questionnaires, process forms and laboratory samples were labelled with matching numbers to maintain confidentiality. Women received support (transport, condoms, family planning visits, selected contraceptive methods, screening of selected Sexual Transmitted Infections (STIs), and reference to appropriate specialized services) to facilitate access to care and treatment programs. A second HIV test was offered to all HIV-negative women at T2.

### Sample size and eligible criteria

Sample size have been estimated in order to rise the number of men partner tested of 10% within women who followed COC (target:15%) compare to women who followed Classical Counselling (CC target: less than 5%), with an alpha type I error of 5% (two-sided test) and a

beta type II risk of 10% (power of 90%). Considered a proportion of 15% of lost to followed-up and inconsistent data, each study site might recruit a minimum of 238 women per group, so 476 women per site [21].

The inclusion criteria consist of the following: (a) to be at least 15 years old, (b) to visit the study site for the first Antenatal Care (ANC-1), (c) have nevered been tested for HIV concerning the current pregnancy, and (d) have a stable partner.

The non-inclusion criteria were: (a) to have a mental deficit at inclusion time, (b) to have a partner absent for more than 6 months in a year, and (c) to be already tested for the current pregnancy.

## Enrolment and randomization

Eligible women and men agreeing to participate had attended a recruitment interview. They were explained the project in more details and were asked to sign the informed consent form.

Women enrolled were randomised to the SC group (no intervention, standard post-test HIV counselling) or the COC group (intervention, couple-oriented post-test HIV counselling). All enrolled women were been given a study card (with project ID number, study group, stages of the study completed) including a ticket for free HIV testing for their partners (funded by the Prenahtest project). Enrolled partners were also been given a project ID number [21].

During the recruitment visit with women, the recruiter had:

- Confirmed the inclusion criteria for women accepting participation to the project and completed the inclusion form

- Contact the local coordinator who had accessed the randomisation list (computerised) and affected the woman to the SC or COC group and affected a study number

- Written the group and study number on the inclusion form and on the woman's study card

- Referred the woman to the interviewer for the T0 inclusion questionnaire. The randomisation list was computer generated.

## Measure of MPI in sexual and reproductive health of the couple

The MPI in sexual and reproductive health of the couple was measured by a composite index variable, built from several variables collected during the study [20, 22, 23]. The variables used to build the MPI index variable were chosen following a literature review. We used the following criteria to define MPI: "*A male partner is involved in the sexual and reproductive health of the couple if he is interested in the different aspects of the preventive and reproductive sexual life of the couple and if he proposes attitudes (communications) and adopt practices which ensure and guarantee the couple's sexual and reproductive health.*" The variables selected to build MPI composite index variable were mainly focused on the attitudes and practices in terms of partner behaviour with regard to the couple's sexual and reproductive health as detailed bellow:

## Attitude / ability

- Risk of HIV infection from male partner

- Partner's risk of HIV infection

- Discussion about contraceptive methods initiated by male partner

- Communication around the condom in couple

- Discussion about HIV in the couple initiated by male partner

- Attendance and discussion about Couple Oriented Counselling (COC)

## Practice

- Male Partner support (consultation, family planning, vaccination)

- Safe sexual intercourse

- Testing of HIV,

- Withdrawal of HIV test results

- Disclosure of the HIV test result with partner

## Statistical analysis

Data entry was done using the EPIINFOS software. Continuous variables were reported by median with Interquartile interval range (25th and 75th percentiles) and by mean and standard deviation, while categorical variables were described as frequencies and percentages. To measure the MPI, we carried out a Multiple Correspondence Analysis (MCA) followed by a mixed classification to affect each participant in a given class according to his behaviour in the different outcomes of the couple's sexual and reproductive health. To study the factors associated with the MPI, we used a logistic regression model for longitudinal data with a GEE approach. We used this method because the outcome was measured at three different time-points coming from the same women so values were correlated and GEE approach was appropriate to get correct standard errors.

The model estimated is given by the equation bellow:

$$\log\left(\frac{P(MPI_{ij} = high)}{1 - P(MPI_{ij} = high)}\right) = \beta_0 + \beta_1 Group + \beta_2 Visit + \beta_3 Group * Visit + \sum_{k=1}^{l} \gamma_k Var_{ijk} + \sum_{k=1}^{l} \theta_k Var_{ik} * Visit + \sum_{k=1}^{l} \delta_k Var_{ik}$$

Where:

$log$ represent the link function;

$i = 1,\ldots,478$ represent the pregnant women enrolled;

$j = 1,2,3$ represent the Visit time attempted by the participant;

$p(MPI_{ij} = high)$ represent the probability for Male Partner's Involvement to be high;

*Group* represent the trial intervention;

*Visit* represent the variable which specify the time of interview during follow-up;

$Var_{ijk}$ represent the explanatory variable number $k$ collected during follow-up (*As explanatory variable we can cite Male partner income activity, gestational age, violence, HIV Status, duration of relationship, religious affiliation, etc.*);

$Var_{ijk}$ represent the explanatory variable number $k$ collected only at enrollment; $\beta_0 \ldots \beta_3, \gamma_k,$ $\theta_k, \delta_k$ are the parameters of the model to be estimated.

According to information criterion (QICu), we chose an exchangeable correlation matrix. We carried out a univariate analysis to select the explanatory variables to be introduced in the initial model of the multivariable analysis in addition to the main exposure factor (post-test counselling group) and potential confounders. The significance threshold considered at this stage was 20%. We used a manual backward stepwise selection method to obtain the final model. To have our final model, we started with the non-significant interaction terms at the 5% threshold (from the least significant) and then with the main terms while controlling for

the confounding biases and looking at the information criterion (QICu). The analysis of the quality fitting of the final model retained was also assessed.

We used R 3.2.3, SAS 9.4 and SPAD 5.5 to carry out our analyses. The association tests (Fisher's exact test) were carried out with R software version 3.2.3.

The MCA and the mixed classification were implemented by SPAD version 5.5 software.

The regression model was estimated by SAS software version 9.4 and the main procedures used to conduct these analyses were as followed:

1. PROC GENMOD with REPEATED option in all the adjustment stages for the logistic regression model with the GEE approach;

2. The option OBSTATS in the specification of the model of the GENMOD procedure allows us to obtain the statistics necessary for the diagnosis of influential and outliers;

The significance threshold considered in our analyses is 5%.

## Ethics statement

The Prenahtest study protocol version 4 of the 18[th] December 2006 received ethical clearance from the National Ethics Committee of Cameroon (Authorization N˚ FWA IRB 00001954) and was registered on Clinical Trials.gov as NCT01494961 [19]. This trial was not registered before the beginning of enrolment because neither the principal investigators, nor the piloting committee thought it was necessary in the context of an intervention trial (not drug). However, this was an oversight on our part and we quickly registered the trial as soon as we realized it. The authors confirm that all ongoing and related trials for this drug/intervention are registered. The privacy of consenting pregnant women and data confidentiality were ensured by the use of ID codes. All participants had signed informed consent without any incentive. HIV tests were offered for free and all women tested positive were placed on ART according to the national guidelines.

## Results

### Characteristics of our study population

A total of 484 pregnant women were screened and 478 were finally enrolled and randomized either to follow CC or COC at ANC-1 (Fig 1). Among them, 73.22% (350/478) had less than 30 years (Table 1). Almost 68% (321/478) of them had a high school level and 64.64% (309/478) were catholic or orthodox christians. In addition, more than half did not have an income activity (55.65%) or did not come to their ANC-1 early (55.23% had a gestational age $\geq 16$ weeks) and less than one third of women were married (27.82%). Furthermore, about two fifths of them were in a fairly long-term relationship (38.49% had spent more than 5 years with their partner), and 21.55% did not knew the level of education of their partner. Their partners were young adults (75.1% were under 40 years) and were catholic or orthodox christians (60.67%). In addition, very few of them had no income activity (9%) and around one third had a university level of education (32.22%).

The distribution of these characteristics according to CC or COC group allowed us to conclude that the randomization was well done as the two groups were very similar on most of these characteristics (Table 1).

The prevalence of HIV within the cohort estimated at the visit during pregnancy was 11.93% (95% Exact confidence interval from Binomial distribution: 9.0% - 15.4%).

**Table 1. Socio-demographics characteristics of couples at inclusion according to the counselling group within the ANRS 12127–12236 trial cohort Prenahtest, Cameroon, 2009–2011; (N = 478).**

| Variables | Overall n (%) | CC* n (%) | COC* n (%) |
|---|---|---|---|
| **Male partner age group (in year)** | | | |
| ≤ 30 | 147 (30.75) | 77 (32.22) | 70 (29.29) |
| 31–39 | 212 (44.35) | 101 (42.26) | 111 (46.44) |
| ≥ 40 | 78 (16.32) | 43 (17.99) | 35 (14.64) |
| Don't know | 41 (8.58) | 18 (7.53) | 23 (9.62) |
| **Women age group (in year)** | | | |
| ≤ 20 | 59 (12.34) | 35 (14.64) | 24 (10.04) |
| 21–30 | 291 (60.88) | 140 (58.58) | 151 (63.18) |
| ≥ 31 | 128 (26.78) | 64 (26.78) | 64 (26.78) |
| **Duration of relationship (in months)** | | | |
| ≤ 24 | 143 (29.92) | 73 (30.54) | 70 (29.29) |
| 25–60 | 151 (31.59) | 76 (31.80) | 75 (31.38) |
| > 60 | 184 (38.49) | 90 (37.66) | 94 (39.33) |
| **Gestational age (in weeks)** | | | |
| < 16 | 208 (43.51) | 103 (43.10) | 105 (43.93) |
| ≥ 16 | 264 (55.23) | 132 (55.23) | 132 (55.23) |
| Don't know | 6 (1.26) | 4 (1.67) | 2 (0.84) |
| **Marital status** | | | |
| Single / Widowed / Divorced | 195 (40.79) | 93 (38.91) | 102 (42.68) |
| Married | 133 (27.82) | 68 (28.45) | 65 (27.20) |
| Free union | 150 (31.38) | 78 (32.64) | 72 (30.13) |
| **Male partner income activity** | | | |
| Yes | 435 (91.00) | 218 (91.21) | 217 (90.79) |
| No | 43 (9.00) | 21 (8.79) | 22 (9.21) |
| **Women income activity** | | | |
| Yes | 212 (44.35) | 102 (42.68) | 110 (46.03) |
| No | 266 (55.65) | 137 (57.32) | 129 (53.97) |
| **Male partner education level** | | | |
| No level or Primary | 24 (5.02) | 13 (5.44) | 11 (4.60) |
| high school | 197 (41.21) | 102 (42.68) | 95 (39.75) |
| University | 154 (32.22) | 75 (31.38) | 79 (33.05) |
| Don't know | 103 (21.55) | 49 (20.50) | 54 (22.59) |
| **Women education level** | | | |
| No level or Primary | 55 (11.50) | 25 (10.46) | 30 (12.55) |
| high school | 321 (67.15) | 156 (65.27) | 165 (69.04) |
| University | 102 (21.34) | 58 (24.27) | 44 (18.41) |
| **Male partner religious affiliation** | | | |
| Catholic / Orthodox Christianity | 290 (60.67) | 145 (60.67) | 145 (60.67) |
| Other Christianity | 119 (24.90) | 59 (24.69) | 60 (25.10) |
| Islam | 26 (5.44) | 15 (6.28) | 11 (4.60) |
| Other | 29 (6.07) | 15 (6.28) | 14 (5.86) |
| Don't know | 14 (2.93) | 5 (2.09) | 9 (3.77) |
| **Women religious affiliation** | | | |
| Catholic / Orthodox Christianity | 309 (64.64) | 142 (59.41) | 167 (69.87) |
| Other Christianity | 143 (29.92) | 84 (35.15) | 59 (24.69) |
| Islam | 19 (3.97) | 10 (4.18) | 9 (3.77) |
| Other | 7 (1.46) | 3 (1.26) | 4 (1.67) |

* CC: Classical Counselling; COC: Couple Oriented Counselling.

## MPI in the sexual and reproductive health of the couple

The result of the MCA followed by mixed classification gave for each time of the study 2 groups of MPI (S1 Table). Overall, for each time, we had a group where the partners did not take an HIV test, did not shared the results of the HIV test with partner, did not initiated discussion about HIV or communication about the condom use in the couple. In addition, in this group, they had not accompanied their partner in prenatal consultation, or did not practiced safe intercourse sex (using a condom when you did not know the status of your partner). This group was characterized as low MPI. The other group in which the partners rather displayed behaviours opposite to the first one was characterized as high MPI. The highly MPI group significantly increased during pregnancy (19.76% vs 11.72%; p-value:0.0005) and then decreased after delivery (19.76% vs 11.85%; p-value:0.0016) (S2 Table).

Furthermore, by looking at the evolution of the high MPI by counselling group (Fig 2), we observed the same trend as in the overall study population: the high involvement increased during pregnancy and then decreased 6 months after delivery in the two post-test counselling groups. However, the decrease seemed to be huge in the CC group (Fig 2). After analysing the influence of lost to follow-up in the evolution of the high MPI (S1 Fig and S1 Result) we concluded that the new biomedical intervention (COC counselling) would seemed to be more effective in the medium or long term than the standard counselling (intention-to-treat analysis: 6 months after pregnancy 14.77% vs 8.82%; p-value:0.0435; during pregnancy 18.54% vs 20.95%; p-value: 0.2683).

## Factors associated to IPV

In multivariable analysis, the post-test counselling group was significantly associated with the evolution of the MPI. On average, after adjusting for other covariates, partners of women who

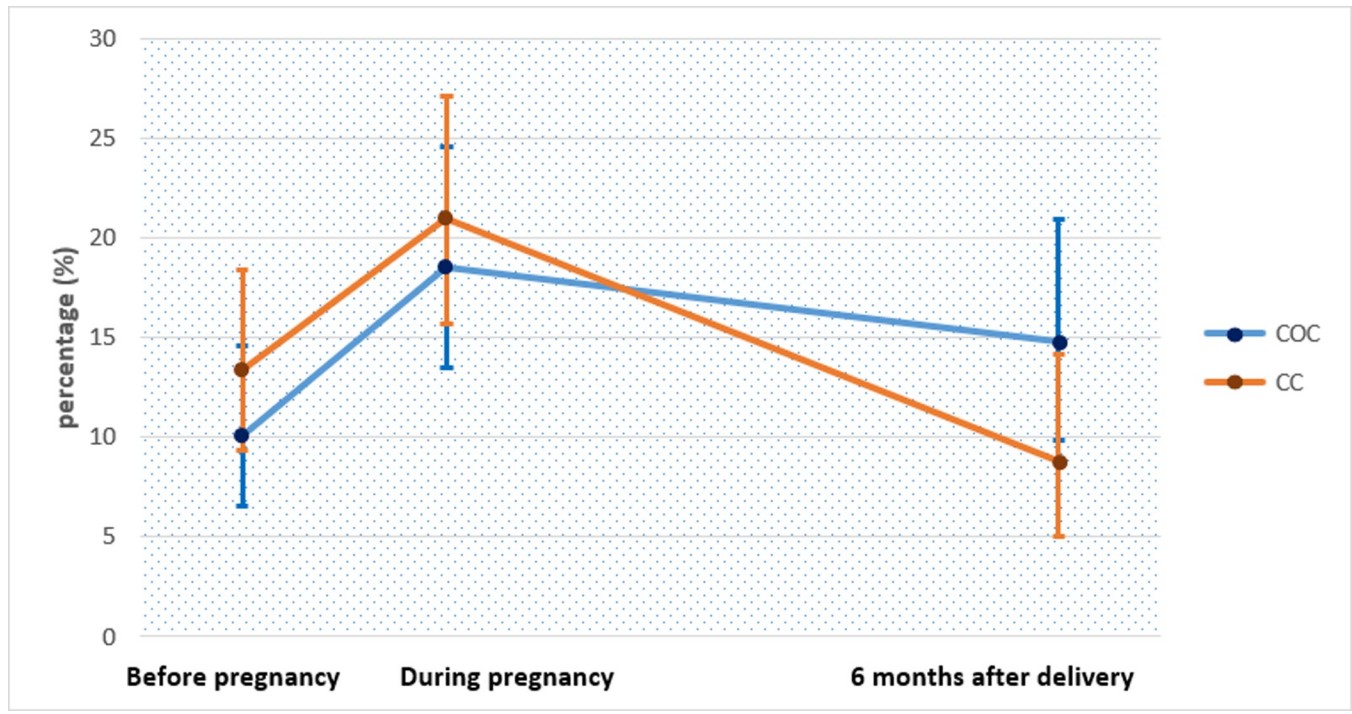

**Fig 2. Evolution of the proportion (%) of high involvement of male partner in sexual and reproductive health of the couple by counselling group among pregnant women cohort, in Yaounde, Cameroon, ANRS 12127-12236/Prenahtest trial (2009–2011).**

followed COC were more likely to be highly involved (aOR: 1.45; 95% CI: 1.00–2.10) (Table 2). However, the probability of being highly involved decreased over time (aOR: 0.77; 95% CI: 0.59–1.00). In addition, partners of women who did not had an income activity or who did not exercised physical or verbal violence in the couple during the follow-up were more likely to be highly involved in sexual and reproductive health of the couple (aOR: 2.90; 95% CI: 1.96–4.29 and aOR: 1.70; 95% CI: 1.07–2.68 respectively). Moreover, the partners of women who came to their ANC-1 at ≤ 16 weeks of pregnancy were 1.4 times more likely to be highly involved than the partners of women who came to their ANC 1 at 16 weeks of pregnancy or more (aOR: 1.37; 95% CI: 1.00–1.89).

## Discussion

The estimated HIV prevalence during pregnancy in our study population was 11.93% (95% Exact confidence interval from Binomial distribution: 9.00% - 15.40%): It was greater than the national HIV prevalence among pregnant women which was estimated at 7.8% during 2009 National sentinel survey [3], which was already high. Other recent studies have estimated that more than 50% of new infections occur within "stable couples (married or cohabiting)" [24, 25].

Our results showed that the new biomedical intervention Couple-Oriented Counselling (COC) improved the high MPI of the partner in the sexual and reproductive health of the couple during follow-up. This interesting result is added to other positive results on the effectiveness of this intervention on other outcomes of sexual and reproductive health such as partner testing or conjugal communication in the couple [26, 27].

In addition, either before or during pregnancy or even after delivery, the proportion of high MPI remains low (less than 20%). This result has already been described in the literature,

**Table 2. Factors associated with the evolution of high MPI in the couple's sexual and reproductive health, ANRS trial 12127–12236 Prenahtest, Cameroon, 2009–2011.**

| Variables | n* | % HMPI** | *Univariable analysis OR (CI at 95%)* | *Multivariable analysis# aOR (CI at 95%)* |
|---|---|---|---|---|
| **Post-test Counselling Group** | | | | |
| CC | 239 | 13.39 | 1 | 1 |
| COC | 239 | 10.04 | 0.49 (0.23–1.04) | 0.46 (0.21–1.03) |
| **Visit** | | | 0.87 (0.66–1.15) | 0.77 (0.59–1.00) |
| **Post-test Counselling Group*Visit** | | | | |
| CC | | | 1 | 1 |
| COC | | | 1.43 (1.00–2.99) | 1.45 (1.00–2.10) |
| **Male partner income activity** | | | | |
| No | 43 | 25.58 | 2.71 (1.85–3.97) | 2.90 (1.96–4.29) |
| yes | 435 | 10.34 | 1 | 1 |
| **Male partner had exercised a type of violence (verbal or physical)** | | | | |
| No | 223 | 13.9 | 1.57 (1.04–2.38) | 1.70 (1.07–2.68) |
| Yes | 255 | 9.8 | 1 | 1 |
| **Gestational age (in weeks)** | | | | |
| <16 | 208 | 15.87 | 2.17 (1.01–4.66) | 1.37 (1.00–1.89) |
| ≥ 16 | 264 | 8.71 | 1 | 1 |

*: Number at inclusion;

** HMPI: Percentage of High MPI at inclusion;

#: Multivariable model included also primiparous and duration of the relationship variables.

where there are proportions of high MPI around 33% [28], around 26% [29–31] or even around 13% [13]. However, our results show that the MPI is higher during pregnancy than before pregnancy or after delivery. This corroborates with the sociocultural context in Cameroon where pregnancy is a period in the couple during which male partner is more involved and woman receives more attention from him especially for young couples (married or in free union) since in most cases they are waiting their first child.

The multivariable analysis modelling revealed that a pregnant woman who had her ANC-1 before the 16th week of pregnancy was associated with the high MPI in the sexual and reproductive health of the couple. This seems entirely plausible because an involved partner should care about the health of his pregnant wife through regular and fairly early monitoring of the pregnancy by health personnel. Similarly, partners with no income activity seemed more highly involved than those who have an income activity and this could be explained by the fact that partners with income activity do not always have the time to discuss properly with their wife about all the outcomes of sexual and reproductive health of the couple or even to be tested for HIV and are generally limited to support financially the needs of his pregnant partner. Moreover, Men's roles are sometimes perceived to be limited to provision of appropriate food and supplies, physical and emotional support. Generally, ANC attendance is considered a woman's private activity because even health care providers are mostly female in many facility as already described [31].

Our results also showed that partners who did not use verbal or physical violence in their relationship were more highly involved in the sexual and reproductive health of the couple during the follow-up, firstly because this created a favourable environment to discussion and communicate on the outcomes of sexual and reproductive health of the couple and secondly demonstrated some kind of attention from the male partner. Some studies published corroborated these results in the literature [32, 33].

In our study we used MCA followed by mixed classification to build MPI. Compared to the manual scoring approach used by some authors to build MPI [29, 30], this technique has the advantage of taking into account the homogeneity of the participants group and the weighting of the different variables used.

The population included in our study lived in urban areas and had an HIV prevalence higher than the national average prevalence. It is therefore difficult to conclude that the COC intervention is effective and integrated into the national health system because we do not know how it would work in rural area or in regions of the country with a low prevalence of HIV. However, the effectiveness of this intervention has already been demonstrated on partner testing in countries with low HIV prevalence such as India and Georgia in the Prenahtest trial [17].

We reported a lost-to-follow-up rate around 25% at the end of the follow-up 6 months after delivery. This may have somehow affected the power of our study. It should be noted, however, that there was no difference between the rates of loss of follow-up in the two post-test counselling groups and that the socio-demographic characteristics of those lost to follow-up were not different.

Moreover, dropping out for a reason which is correlated with MPI in any intervention group could have introduced potential bias into the analysis. In our work we did not assessed the impact of potential bias due to informative censoring.

The fact of having used the data from the Prenahtest project which was not designed with aim of responding primary to our study objective also limited the choice of some explanatory variables, in particular on the financial support of the partner.

## Conclusion

Our study found that the proportion of high MPI in the couple's sexual and reproductive health remains low. It varied between 10% and 20%. After being effective on partner testing

and marital communication, our results also shown that COC is efficient in improving the MPI in the couple's sexual and reproductive health. This allows us to consider the proposal of COC as a strategy or tool with the potential to strengthen prevention among "stable" couples in the fight against the HIV/AIDS epidemic. Barriers to high MPI identified in this study should support the improvement of sensitization messages to "stable" couples specifically.

## Supporting information

**S1 Fig. Evolution of the proportion (%) of high involvement of male partner in sexual and reproductive health of the couple by counselling group among pregnant women of the cohort who attempted all visit, in Yaounde, Cameroon, ANRS 12127-12236/Prenahtest trial (2009–2011).**
(TIF)

**S2 Fig. Evolution of the proportion (%) of high involvement of male partner in sexual and reproductive health of the couple by counselling group among pregnant women of the cohort who attempted all visit and the male partner was not involved at ANC-1, in Yaounde, Cameroon, ANRS 12127-12236/Prenahtest trial (2009–2011).**
(TIF)

**S1 Table. Description of the MPI clusters before pregnancy, during pregnancy and six months after delievery obtained from mixed classification, Prenahtest ANRS 12127–12236 Prenahtest, Cameroon, 2009–2011.**
(PDF)

**S2 Table. Description of the MPI clusters from mixed classification during follow-up, Prenahtest ANRS 12127–12236 Prenahtest, Cameroon, 2009–2011.**
(PDF)

**S1 Result. Sensitive analysis of the influence of lost to follow-up in the evolution of the high MPI, Prenahtest ANRS 12127–12236 Prenahtest, Cameroon, 2009–2011.**
(PDF)

**S1 Checklist. CONSORT 2010 checklist of information to include when reporting a randomised trial**[*]**.**
(PDF)

## Acknowledgments

Prenahtest trial was sponsored by the French Agency ANRS and by EGPAF. The authors thank all pregnant women who accepted to participate in this study; they also thank the Centre Mere-Enfant. The contributions of Tatiana Mossus, Denise Amassana and other members of the Prenahtest team in Yaounde are highly appreciated. The authors also acknowledge the entire Prenahtest team: Marija Miric and Eddy Perez-Then from CENISMI (Santo Domingo, Dominican Republic); Maia utsashvili, Maia Kajaia, George Kamkamidze from Maternal Child Care Union (Tbilisi, Georgia); Shrinivas Darak and Sanjeevani Kulkarni from Prayas Health Group (Pune, India); Fred Eboko from UMR912 INSERM-IRD(Marseille, France); Annabel Desgrees du Lou from UMR 196 CEPED (Paris, France).

Thank you to Brigitte Bazin, Claire Rekacewiz, Laurence Quinty (ANRS) and Catherine Wilfert (EGPAF) for encouraging the study team through the trial.

Special Thanks to Bernard Chawo Silenou (Helmholtz Centre for Infection Research, Department of Epidemiology, Braunschweig, Germany) for edited our article for language.

## Author Contributions

**Conceptualization:** Mathurin Cyrille Tejiokem, Joanna Orne-Gliemann, Paul Koki Ndombo, Anne Cécile Bissek, Patrice T. Tchendjou.

**Data curation:** Cyprien Kengne-Nde, Simon Cauchemez, Patrice T. Tchendjou.

**Formal analysis:** Cyprien Kengne-Nde, Simon Cauchemez.

**Funding acquisition:** Joanna Orne-Gliemann, Patrice T. Tchendjou.

**Investigation:** Cyprien Kengne-Nde, Mathurin Cyrille Tejiokem, Joanna Orne-Gliemann, Bernard Melingui, Paul Koki Ndombo, Anne Cécile Bissek, Patrice T. Tchendjou.

**Methodology:** Cyprien Kengne-Nde, Mathurin Cyrille Tejiokem, Joanna Orne-Gliemann, Ngo A. Essounga, Anne Cécile Bissek, Patrice T. Tchendjou.

**Project administration:** Mathurin Cyrille Tejiokem, Joanna Orne-Gliemann, Patrice T. Tchendjou.

**Resources:** Joanna Orne-Gliemann, Simon Cauchemez, Patrice T. Tchendjou.

**Software:** Cyprien Kengne-Nde, Simon Cauchemez.

**Supervision:** Cyprien Kengne-Nde, Joanna Orne-Gliemann, Bernard Melingui, Paul Koki Ndombo, Ngo A. Essounga, Anne Cécile Bissek, Patrice T. Tchendjou.

**Validation:** Mathurin Cyrille Tejiokem, Joanna Orne-Gliemann, Paul Koki Ndombo, Ngo A. Essounga, Anne Cécile Bissek, Simon Cauchemez, Patrice T. Tchendjou.

**Visualization:** Cyprien Kengne-Nde, Simon Cauchemez, Patrice T. Tchendjou.

**Writing – original draft:** Cyprien Kengne-Nde.

**Writing – review & editing:** Cyprien Kengne-Nde, Mathurin Cyrille Tejiokem, Joanna Orne-Gliemann, Bernard Melingui, Paul Koki Ndombo, Ngo A. Essounga, Anne Cécile Bissek, Simon Cauchemez, Patrice T. Tchendjou.

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
