## [Decision Letter · Decision Letter 0]

29 Mar 2021

PONE-D-20-32163

Couple Oriented Counselling improves male partner involvement in sexual and reproductive health of the couple: evidence from the ANRS 12127/12236 PRENAHTEST Cohort in Cameroon.

PLOS ONE

Dear Dr. Kengne-Nde,

Thank you for submitting your manuscript to PLOS ONE. After careful consideration, we feel that it has merit but does not fully meet PLOS ONE’s publication criteria as it currently stands. Therefore, we invite you to submit a revised version of the manuscript that addresses the points raised during the review process.

The manuscript has been evaluated by three reviewers, and their comments are available below. You will see the reviewers have commented on the interest of your manuscript. However, the reviewers have also raised concerns and the manuscript will need significant revision before it can be considered for publication – you should anticipate that the reviewers will be re-invited to assess the revised manuscript, so please ensure that your revision is thorough. I have outlined some of the key concerns noted by the reviewers below, but you should respond to all concerns mentioned by the reviewers in your response-to-reviewers document. 

The key concerns noted by the reviewers relate to the description of the participant enrollment, randomization, and allocation procedures; as well as the discussion of the results in light of the original primary outcomes and power calculations. These issues impact the interpretation of the results and should be explored.

We look forward to receiving your revised manuscript.

Kind regards,

Danielle Poole

Staff Editor

PLOS ONE

Journal Requirements:

3. Thank you for submitting your clinical trial to PLOS ONE and for providing the name of the registry and the registration number. The information in the registry entry suggests that your trial was registered after patient recruitment began. PLOS ONE strongly encourages authors to register all trials before recruiting the first participant in a study.

a) your reasons for your delay in registering this study (after enrolment of participants started);

b) confirmation that all related trials are registered by stating: “The authors confirm that all ongoing and related trials for this drug/intervention are registered”.

"This work was supported by the Agence Nationale de Recherches sur le SIDA et les hépatites

virales (French National Agency on AIDS Research) (grant ANRS 12127). Complementary

funding was provided by the Elizabeth Glaser Pediatric AIDS Foundation (Sub-agreement

354–07). No funding bodies had any role in study design, data collection and analysis,

decision to publish, or preparation of the manuscript."

"The funders had no role in study design, data collection and analysis, decision to

publish, or preparation of the manuscript."

5, Please include captions for your Supporting Information files at the end of your manuscript, and update any in-text citations to match accordingly. Please see our Supporting Information guidelines for more information: http://journals.plos.org/plosone/s/supporting-information.

Reviewers' comments:

Reviewer's Responses to Questions

**Comments to the Author**

1. Is the manuscript technically sound, and do the data support the conclusions?

Reviewer #1: Partly

Reviewer #2: Yes

Reviewer #3: Partly

2. Has the statistical analysis been performed appropriately and rigorously? 

Reviewer #1: No

Reviewer #2: I Don't Know

Reviewer #3: I Don't Know

3. Have the authors made all data underlying the findings in their manuscript fully available?

Reviewer #1: Yes

Reviewer #2: Yes

Reviewer #3: Yes

4. Is the manuscript presented in an intelligible fashion and written in standard English?

Reviewer #1: Yes

Reviewer #2: Yes

Reviewer #3: No

5. Review Comments to the Author

Reviewer #1: This is a dated, although interesting, report of the ANRS 12127-Prenahtest trial conducted to evaluate the impact of Couple Oriented Counseling (COC) for increasing the prevalence of male partner involvement (MPI) in women attending antenatal clinics in Cameroon. There are several aspects of the study design and of the statistical analysis that need to be clarified or improved.

1. Randomisation. There is no description of how the randomisation schedule was produced. I assume standard randomisation was employed but needs to be described (random numbers, computer generated etc).

2. Concealment of allocation. Similarly it is not stated whether the randomisation schedule was concealed to the trial staff who recruited the women in the trial.

3. Outcome. The trial was powered in order to detect a 10% increase in the rate of HIV testing in the partners of women receiving COC vs. CC. However, the intervention was then evaluated using a completely different outcome. Why was this? Are results confirmed using the primary outcome used for the power calculations? Also what was the expected underlying prevalence of testing in the CC group? That would have affected the power calculations.

4. Analysis1. The results of the unsupervised analysis should be shown in graphical way to illustrate the clustering of the questions with regards of the classification into low and high MPI. This could go as a single Supplementary Figure showing the first two principal components (replacing Tables S3-S5 which are difficult to follow).

5. Analysis 2. It is a randomised trial and Table 1 shows that randomisation has worked well. The description of the logistic model in confusing. For the evaluation of the effect of the intervention (COC vs CC) there is no need to control for confounding variables as confounding bias is minimised by design. Because there was a marked proportion of women who have been lost to follow-up, one thing that could be done was an adjustment for potential informative censoring using inverse probability of censoring weights. Regarding the association between other factors and the risk of high level MPI the GEE logistic regression is reasonable. However, authors should clarify that this was done because the outcome was measured at 3 different time-points coming from the same women so values are correlated and GEE are needed to get correct standard errors. In contrast, the key exposures of interest (e.g. MP participation at early weeks of pregnancy and income level of the partner) appear to be variables that are unlikely to vary over the study period so very little is added by using repeated measurements of these factors. Because women were not randomised to levels of these factors makes sense to be concerned about confounding in this case. Nevertheless, the construction of the model was derived using an automatic stepwise procedure which should be avoided outside of the context of prediction. Suggest that the analysis is restricted to three GEE logistic regression models: 1) effect of intervention (COC vs CC), only univariable, report OR with 95% CI as Table 2, not in supplementary tables; 2) effect of MP participation at early weeks of pregnancy measured at T0, univariable model and model adjusted for key confounders for this specific association. This should be decided on the basis of previous literature or axiomatic knowledge; 3) effect of partner income level measured at T0, univariable model and model adjusted for key confounders for this specific association. This also should be decided on the basis of previous literature or axiomatic knowledge (of note they might be different factors compared to model 2). Results of analyses 2) and 3) should be shown in a separate Table 3, not Supplementary.

6. Analysis 3. There appears to be interaction between intervention and time (effect larger during pregnancy compared to at the beginning of gestational period or after delivery). This is shown in a number of Figures but need to be formally tested using an interaction test in the GEE model.

Other points

Some sentences are unclear and there are several typos and word spelling (e.g. analyzes?) than need to be corrected. Please see the list below

Abstract conclusions. Our results also confirm that strengthening outreach towards "stable" couples and address barriers could go a long way to improve PMTCT outcomes in Cameroon. Meaning of the sentence unclear

Page 4. Behavioral studies suggest an improvement in the content of counseling for HIV testing and an approach favoring greater MPI. Content or context?

Page 5. For women with personalized information, it provides as well as tools and strategies to actively involve her partner within the prenatal HIV counseling and testing process. Meaning of the sentence is unclear

A few lines below, typo conducted, not conducetc

Page 6. The MPI in sexual and reproductive health of the couple was measure by a composite index variable. Typo was measured, not measure

Page 7. The description of the model using the mathematical formula is too technical for the journal. I would move it to the Appendix

 represent the post-test counseling group (Couple Oriented Counseling or Classical Counseling); Assume the author refers to post HIV testing. This is confusing I would say Group represents the trial intervention

Page 9. In addition, very few of them had no income activity (9%) and around one third had a higher level of education (32.22%). Higher than what?

Page 10. On average, adjusted to other covariates... Replace with ‘On average, after adjusting for other covariates…’

Page 11. This corroborates with the sociocultural context in Cameroon where pregnancy is a period in the couple during which male partner is more involve. Typo involved, not involve

Reviewer #2: The paper is interesting even if the data are old.

Some suggestions

Methods:

1) Please specify the references related to the counseling modeling and to the Health Belief Model (pag 5)

2) I don't understand as the enrolment occured before HIV testing: please specify

3) Please specify the literature used to define the MPI index

Discussion:

It would be interesting to deepen the aspects related to the sociocultural context in Cameroon concerning the pregnancy period.

Reviewer #3: Thank you for this manuscript which adds valuable information to the literature available.

Introduction

noted that authors used 2018 statistics on HIV and AIDS and I suggest that they update that information

The sentence about EID seems to be a statement that is not supported.

Add brief literature on Couple Oriented Counselling in the introduction

Methods

Add in the reference regarding COC

Write the sample size section in tense

Results

The last sentence under the MPI in the sexual and reproductive health of the couple is more a premature discussion of the results.

Discussion

The first paragraph should summarise the key findings while answeing the objectives of the study. Then state the key findings so that a reader is aware of the results that will be discussed.

then present the finding then support it with evidence with literature that is similar to the findings , or different and if different provide the evidence that explains the alternative views.

The 4th and 5th paragraphs have not been situated in literature so revise those

the 6th paragraph is more appropriate for the methods section

The article should be edited for language

6. PLOS authors have the option to publish the peer review history of their article (what does this mean?). If published, this will include your full peer review and any attached files.

Reviewer #1: No

Reviewer #2: No

Reviewer #3: **Yes: **Alinane Linda Nyondo-Mipando

---

## [Author Response · Author response to Decision Letter 0]

31 May 2021

Editor

Authors: Our manuscript meets PLOS ONE's style requirements, including those for file naming Thank you.

2. We suggest you thoroughly copyedit your manuscript for language usage, spelling, and grammar. If you do not know anyone who can help you do this, you may wish to consider employing a professional scientific editing service. Whilst you may use any professional scientific editing service of your choice, PLOS has partnered with both American Journal Experts (AJE) and Editage to provide discounted services to PLOS authors. Both organizations have experience helping authors meet PLOS guidelines and can provide language editing, translation, manuscript formatting, and figure formatting to ensure your manuscript meets our submission guidelines. To take advantage of our partnership with AJE, visit the AJE website (http://learn.aje.com/plos/) for a 15% discount off AJE services. To take advantage of our partnership with Editage, visit the Editage website (www.editage.com) and enter referral code PLOSEDIT for a 15% discount off Editage services. If the PLOS editorial team finds any language issues in text that either AJE or Editage has edited, the service provider will re-edit the text for free.

Authors: We had copyedited our revised manuscript by one of our native English speaker. Find below his information.

Name: Bernard Chawo Silenou 

Institute: Helmholtz Centre for Infection Research, Department of Epidemiology, Braunschweig, Germany) 

Address: Inhoffenstraße 7 | 38124 Braunschweig |

Mail: Bernard.Silenou@helmholtz-hzi.de

We also resubmitted a copy of our manuscript edited using track changes, and a clean copy of the edited manuscript. Thank you.

3. Thank you for submitting your clinical trial to PLOS ONE and for providing the name of the registry and the registration number. The information in the registry entry suggests that your trial was registered after patient recruitment began. PLOS ONE strongly encourages authors to register all trials before recruiting the first participant in a study.

a) your reasons for your delay in registering this study (after enrolment of participants started);

b) confirmation that all related trials are registered by stating: “The authors confirm that all ongoing and related trials for this drug/intervention are registered”.

Authors: The current version of our manuscript has the reasons for delay in registering our study in the “Ethics Statement section” as well as the statement “The authors confirm that all ongoing and related trials for this drug/intervention are registered”. Thank you.

"This work was supported by the Agence Nationale de Recherches sur le SIDA et les hépatites virales (French National Agency on AIDS Research) (grant ANRS 12127). Complementary funding was provided by the Elizabeth Glaser Pediatric AIDS Foundation (Sub-agreement 354–07). No funding bodies had any role in study design, data collection and analysis, decision to publish, or preparation of the manuscript."

"The funders had no role in study design, data collection and analysis, decision to

publish, or preparation of the manuscript."

Authors: We removed any funding-related text from the manuscript. We would like our Funding Statement reads as follows: 

“This study was supported by the Agence Nationale de Recherches sur le SIDA et les hépatites virales (French National Agency on AIDS Research) (grant ANRS 12127). Complementary funding was provided by the Elizabeth Glaser Pediatric AIDS Foundation (Sub-agreement 354–07). The funders had no role in study design, data collection and analysis, decision to publish, or preparation of the manuscript”.

Thank you.

Authors: We have included captions for our Supporting Information files at the end of the manuscript and updated any in-text citations to match accordingly. Thank you.

Reviewer #1:

This is a dated, although interesting, report of the ANRS 12127-Prenahtest trial conducted to evaluate the impact of Couple Oriented Counseling (COC) for increasing the prevalence of male partner involvement (MPI) in women attending antenatal clinics in Cameroon. There are several aspects of the study design and of the statistical analysis that need to be clarified or improved.

Authors: We thank the Reviewer 1 for this summary.

1. Randomisation. There is no description of how the randomisation schedule was produced. I assume standard randomisation was employed but needs to be described (random numbers, computer generated etc).

Authors: Thank you for your comment which gives us the opportunity to provide more details on enrolment and randomization. Effectively the randomisation list had be computer generated. Indeed, we have created a section title "Enrolment and randomization" at page 6. Thank you.

2. Concealment of allocation. Similarly, it is not stated whether the randomisation schedule was concealed to the trial staff who recruited the women in the trial.

Authors: The randomisation schedule was concealed to the trial staff who recruited the women in the trial. Indeed, after confirmed inclusion criteria, when women accepted to participate, the recruiter had to contact the local coordinator who had to access the randomisation list (computerised) and affected the woman to the SC or COC group and affected a study number. These details have been added under "Enrolment and randomization" sub-section at page 6 Thank you.

3. Outcome. The trial was powered in order to detect a 10% increase in the rate of HIV testing in the partners of women receiving COC vs. CC. However, the intervention was then evaluated using a completely different outcome. Why was this? Are results confirmed using the primary outcome used for the power calculations? Also, what was the expected underlying prevalence of testing in the CC group? That would have affected the power calculations.

Authors: The intervention was already evaluated using the primary outcome which to increase at least 10% the rate of HIV testing in the partner of women receiving COC. This was already published. Please find below the published paper on this and the link [1]. Our study evaluated this intervention on a secondary outcome. The underlying prevalence of testing in the CC group expected was less than 5%. We add these details in the current version of the paper under “Sample size and Eligible criteria” sub-section at page 6.

[1] J. Orne-Gliemann et al., « Increasing HIV testing among male partners », AIDS, vol. 27, no 7, Art. no 7, avr. 2013, doi: 10.1097/QAD.0b013e32835f1d8c.

Thank you.

4. Analysis1. The results of the unsupervised analysis should be shown in graphical way to illustrate the clustering of the questions with regards of the classification into low and high MPI. This could go as a single Supplementary Figure showing the first two principal components (replacing Tables S3-S5 which are difficult to follow).

Authors: Thank you to the reviewer 1 for this suggestion. Indeed, as the representation of the first two factorial axes explains only a part of the total inertia, we preferred to keep the results of the mixed classification which was carried out on the total initial at each time point. The tables S3-S5 were condensed in one table to allow a good readability. Thank you.

5. Analysis 2. It is a randomised trial and Table 1 shows that randomisation has worked well. The description of the logistic model in confusing. For the evaluation of the effect of the intervention (COC vs CC) there is no need to control for confounding variables as confounding bias is minimised by design. Because there was a marked proportion of women who have been lost to follow-up, one thing that could be done was an adjustment for potential informative censoring using inverse probability of censoring weights. Regarding the association between other factors and the risk of high level MPI the GEE logistic regression is reasonable. However, authors should clarify that this was done because the outcome was measured at 3 different time-points coming from the same women so values are correlated and GEE are needed to get correct standard errors. In contrast, the key exposures of interest (e.g. MP participation at early weeks of pregnancy and income level of the partner) appear to be variables that are unlikely to vary over the study period so very little is added by using repeated measurements of these factors. Because women were not randomised to levels of these factors makes sense to be concerned about confounding in this case. Nevertheless, the construction of the model was derived using an automatic stepwise procedure which should be avoided outside of the context of prediction. Suggest that the analysis is restricted to three GEE logistic regression models: 1) effect of intervention (COC vs CC), only univariable, report OR with 95% CI as Table 2, not in supplementary tables; 2) effect of MP participation at early weeks of pregnancy measured at T0, univariable model and model adjusted for key confounders for this specific association. This should be decided on the basis of previous literature or axiomatic knowledge; 3) effect of partner income level measured at T0, univariable model and model adjusted for key confounders for this specific association. This also should be decided on the basis of previous literature or axiomatic knowledge (of note they might be different factors compared to model 2). Results of analyses 2) and 3) should be shown in a separate Table 3, not Supplementary.

Authors: Thank you to reviewer 1 for this consistent comment on the methodology used in our paper. As you noted we had a marked proportion of women who have been lost to follow-up. We think that the suggestion for an adjustment for potential informative censoring using inverse probability of censoring weights would not be really necessary since in our design the sample size had taken into consideration a proportion of lost to follow-up. And moreover, there was no difference between the rates of loss of follow-up in the two intervention groups and the socio-demographic characteristics of those lost to follow-up were not different to the participant who participated up to the end of the study. We have adjusted the sub-section "Statistical analysis" to explain the reason for our choice of the logistic regression model with the GEE approach on page 8. Thanks to the reviewer for this observation. Indeed, we used a manual backward stepwise selection approach and not an automatic one, and this was not well specified before. This has been adjusted in the current version of the manuscript. For the model suggestions proposed by reviewer 1, we believe that these elements have already been taken into account in some way. Indeed, for the proposed GEE 1) model, the "Univariable analysis" part of Table 2 is already reporting this information. For the proposed models 2) and 3) the model presented in the current version have used the previous literature or axiomatic knowledge to adjust the model for key confounders. In addition, the current model uses variables such as "partner income level" measured at the three time points to avoid any possible bias in the estimation of the association with MPI even if this would be marginal. We therefore think that it could be more valuable to keep the estimated model reported in table 2. We would like to thank once more the reviewer 1 for these remarks.

Thank you.

6. Analysis 3. There appears to be interaction between intervention and time (effect larger during pregnancy compared to at the beginning of gestational period or after delivery). This is shown in a number of Figures but need to be formally tested using an interaction test in the GEE model.

Authors: the interaction between intervention and time was tested in the GEE model (Variable “Post-test Counseling Group*Visit”) and was significant. Indeed, partners of women who followed COC were more likely to be highly involved as time increased. Thank you.

Other points 

Some sentences are unclear and there are several typos and word spelling (e.g. analyzes?) than need to be corrected. Please see the list below 

Authors: We have corrected these typos. Thank you.

Abstract conclusions. Our results also confirm that strengthening outreach towards "stable" couples and address barriers could go a long way to improve PMTCT outcomes in Cameroon. Meaning of the sentence unclear 

Authors: We divided this sentence in two sentences to make it clearer. Thank you.

Page 4. Behavioral studies suggest an improvement in the content of counseling for HIV testing and an approach favoring greater MPI. Content or context?

Authors: It is context instead of content. We have corrected this in the current version of the manuscript. Thank you.

Page 5. For women with personalized information, it provides as well as tools and strategies to actively involve her partner within the prenatal HIV counseling and testing process. Meaning of the sentence is unclear A few lines below, typo conducted, not conducetc 

Authors: We have cancelled this sentence and replace with others sentences to make this clearer? We have also corrected the typo conducted instead of conductec . Thank you.

Page 6. The MPI in sexual and reproductive health of the couple was measure by a composite index variable. Typo was measured, not measure

Authors: We have corrected the typo measure by measured . Thank you.

Page 7. The description of the model using the mathematical formula is too technical for the journal. I would move it to the Appendix 

represent the post-test counseling group (Couple Oriented Counseling or Classical Counseling); Assume the author refers to post HIV testing. This is confusing I would say Group represents the trial intervention

Authors: Thank you to reviewer 1 for this suggestion. We think it is important to keep the description of the model in this sub-section for a good understanding of the steps of the analysis strategy described below. Moreover, readers with a background in biostatics could easily appropriate the static model for its eventual reproduction in other study. 

We have replaced Group definition as suggested by the reviewer.

Thank you.

Page 9. In addition, very few of them had no income activity (9%) and around one third had a higher level of education (32.22%). Higher than what?

Authors: We meant university level of education. This have been corrected in the current version of the manuscript. Thank you.

Page 10. On average, adjusted to other covariates... Replace with ‘On average, after adjusting for other covariates…’

Authors: We have adjusted this sentence as suggested by the reviewer. Thank you.

Page 11. This corroborates with the sociocultural context in Cameroon where pregnancy is a period in the couple during which male partner is more involve. Typo involved, not involve

Authors: We have corrected as suggested by the reviewer. Thank you.

Reviewer #2: 

The paper is interesting even if the data are old.

Authors: We thank the Reviewer 2 for this appreciation.

Some suggestions

Methods:

1) Please specify the references related to the counseling modeling and to the Health Belief Model (page 5)

Authors: The references related to the counseling modeling and to the Health Belief Model have been specified at page 5. Thank you.

2) I don't understand as the enrolment occurred before HIV testing: please specify

Authors: Eligible women and men agreeing to participate had attended a recruitment interview. They were explained the project in more details and were asked to sign the informed consent form. Women enrolled were randomised to the SC group (no intervention, standard post-test HIV counselling) or the COC group (intervention, couple-oriented post-test HIV counselling). All enrolled women were been given a study card (with project ID number, study group, stages of the study completed) including a ticket for free HIV testing for their partners (funded by the Prenahtest project). Enrolled partners were also been given a project ID number. We have specified these details under subsection “Enrolment and randomization” added in the current version of the manuscript. Thank You.

3) Please specify the literature used to define the MPI index

Authors: We have specified the literature used to define the MPI index at page 6. Thank you

Discussion:

It would be interesting to deepen the aspects related to the sociocultural context in Cameroon concerning the pregnancy period.

Authors: We tried to deepen this aspect related to the sociocultural context in Cameroon concerning the pregnancy period. Thank you.

Reviewer #3: 

Thank you for this manuscript which adds valuable information to the literature available.

Authors: We thank the Reviewer 3 for this appreciation.

Introduction 

noted that authors used 2018 statistics on HIV and AIDS and I suggest that they update that information

Authors: We have updated with 2019 statistics on HIV and AIDS available at the Unaids website. Thank you.

The sentence about EID seems to be a statement that is not supported.

Authors: the sentence about EID is supported with the Camerron’s Statistic on HIV[2]. Please find below the link to access directly to the unaids webside. 

2. Cameroon [Internet]. [cité 29 mai 2021]. Available on:https://www.unaids.org/en/regionscountries/countries/cameroon

Thank you..

Add brief literature on Couple Oriented Counselling in the introduction

Authors: To the best of our knowledge the Couple Oriented Counselling post-test as implemented in our project had not been described before the implementation of our project in the literature. However, we describe it under methods section under “ANRS 12127 - Prenahtest Trial, study site and data collection sub-section”. And moreover the 3rd 4th 5th paragraphs give a brief literature on male participation its benefits on PMTCT services packages. Thank you.

Methods

Add in the reference regarding COC

Authors: We have added reference regarding COC (reference N°20 and N°21). Thank you

Write the sample size section in tense

Authors: We do not understand the suggestion of the reviewer. Please could you specify the tense you are referring to. Thank you. 

Results

The last sentence under the MPI in the sexual and reproductive health of the couple is more a premature discussion of the results.

Authors: Thank you to reviewer 3 for this opinion. We think it would be better for the readers' understanding to keep this sentence here as it explains an analysis done on the influence of lost to follow up participants on the evolution of high male partner involvement and COC efficiency. In addition, there are results (figures to be presented and interpreted).

Discussion

The first paragraph should summarise the key findings while answeing the objectives of the study. Then state the key findings so that a reader is aware of the results that will be discussed. then present the finding then support it with evidence with literature that is similar to the findings, or different and if different provide the evidence that explains the alternative views.,The 4th and 5th paragraphs have not been situated in literature so revise those 

Authors: We have situated The 4th and 5th paragraphs in literature, Thank you.

the 6th paragraph is more appropriate for the methods section

Authors: Thanks to the reviewer 3 for this opinion. We already had described these methods in the methods section. Here we have compared them to other methods used for the same purposes and show the advantages of the ones we used. Thank you

The article should be edited for language

Authors: Thank you to the reviewer 3 for this suggestion. The article has been edited for language.

---

## [Decision Letter · Decision Letter 1]

22 Jun 2021

PONE-D-20-32163R1

Couple oriented counselling improves male partner involvement in sexual and reproductive health of a couple: Evidence from the ANRS PRENAHTEST randomized trial.

PLOS ONE

Dear Dr. Kengne-Nde,

Thank you for submitting your manuscript to PLOS ONE. After careful consideration, we feel that it has merit but does not fully meet PLOS ONE’s publication criteria as it currently stands. Therefore, we invite you to submit a revised version of the manuscript that addresses the points raised during the review process.

We look forward to receiving your revised manuscript.

Kind regards,

Catherine E Oldenburg

Academic Editor

PLOS ONE

Journal Requirements:

Additional Editor Comments (if provided):

Please carefully consider Reviewer 1's comment about analyses for selection bias and consider including a sensitivity analysis using methods are Reviewer 1 notes that would assume data are missing at random as a robustness check for potential selection bias due to the high loss to follow-up.

Reviewers' comments:

Reviewer's Responses to Questions

**Comments to the Author**

1. If the authors have adequately addressed your comments raised in a previous round of review and you feel that this manuscript is now acceptable for publication, you may indicate that here to bypass the “Comments to the Author” section, enter your conflict of interest statement in the “Confidential to Editor” section, and submit your "Accept" recommendation.

Reviewer #1: (No Response)

Reviewer #2: All comments have been addressed

2. Is the manuscript technically sound, and do the data support the conclusions?

Reviewer #1: Partly

Reviewer #2: Yes

3. Has the statistical analysis been performed appropriately and rigorously? 

Reviewer #1: No

Reviewer #2: I Don't Know

4. Have the authors made all data underlying the findings in their manuscript fully available?

Reviewer #1: Yes

Reviewer #2: Yes

5. Is the manuscript presented in an intelligible fashion and written in standard English?

Reviewer #1: Yes

Reviewer #2: Yes

6. Review Comments to the Author

Reviewer #1: The authors did a good job at addressing most of my earlier concerns. However, with regards to my original point #5, I still think that the authors' response is not satisfactory.

R: Using inverse probability of censoring weights would not be really necessary since in our design the sample size had taken into consideration a proportion of lost to follow-up.

*The fact that sample size was adjusted because loss to follow-up was expected guarantees that the power of the study was retained but it does not eliminate the fact that dropping out for a reason which is correlated with the outcome could have introduced bias into the analysis

R: And moreover, there was no difference between the rates of loss of follow-up in the two intervention groups and the socio-demographic characteristics of those lost to follow-up were not different to the participant who participated up to the end of the study.

*Similarly here, even with the same exact incidence of drop out by the intervention group, bias could be introduced if the reason for dropping out was different by group (i.e. correlated with MPI in one group but not in the other, for example if women dropped out from CoC because the partner interfered with that decision)

If authors are reluctant to properly look into the impact of potential bias due to informative censoring they should at least add a limitation in the Discussion

Please also note some suboptimal English in some of the revised parts

"Enrolment and randomization" sub-section at page 6

The randomisation list had be computer generated - The randomisation list WAS computer generated?

page 13. waiting most of the case their first child - in most cases they are waiting....

Reviewer #2: (No Response)

7. PLOS authors have the option to publish the peer review history of their article (what does this mean?). If published, this will include your full peer review and any attached files.

Reviewer #1: No

Reviewer #2: No

---

## [Author Response · Author response to Decision Letter 1]

27 Jun 2021

Editor

Authors: Thank you to Editor for your comment. As requested by reviewer 3, we have just updated reference 1 which referenced the UNAIDS statistics below:

Global HIV & AIDS statistics — 2020 fact sheet [Internet]. [cité 29 mai 2021]. Disponible sur: https://www.unaids.org/en/resources/fact-sheet

The previous one was dated March 2020 and this one is dated May 2021. We forgot to remove the previous one. It has been removed in the current version of the manuscript and the entire bibliography has been updated. Thank you

Additional Editor Comments (if provided):

Please carefully consider Reviewer 1's comment about analyses for selection bias and consider including a sensitivity analysis using methods are Reviewer 1 notes that would assume data are missing at random as a robustness check for potential selection bias due to the high loss to follow-up.

Authors: Thanks to the editor for this comment. We have considered the comments of reviewer 1 and have opted for his proposal to include in the limitation of our work (page 13) that we did not evaluated the impact of potential bias due to informative censoring. Thank you.

Reviewer #1:

The authors did a good job at addressing most of my earlier concerns. 

Authors: We thank the Reviewer 1 for this summary.

However, with regards to my original point #5, I still think that the authors' response is not satisfactory.

R: Using inverse probability of censoring weights would not be really necessary since in our design the sample size had taken into consideration a proportion of lost to follow-up.

*The fact that sample size was adjusted because loss to follow-up was expected guarantees that the power of the study was retained but it does not eliminate the fact that dropping out for a reason which is correlated with the outcome could have introduced bias into the analysis

R: And moreover, there was no difference between the rates of loss of follow-up in the two intervention groups and the socio-demographic characteristics of those lost to follow-up were not different to the participant who participated up to the end of the study.

*Similarly here, even with the same exact incidence of drop out by the intervention group, bias could be introduced if the reason for dropping out was different by group (i.e. correlated with MPI in one group but not in the other, for example if women dropped out from CoC because the partner interfered with that decision) If authors are reluctant to properly look into the impact of potential bias due to informative censoring they should at least add a limitation in the Discussion

Authors: We thank reviewer 1 for this additional explanation which helps us to understand his point of view. We have added this limitation in the discussion (second last paragraph of the discussion, page 13). Thank You

Please also note some suboptimal English in some of the revised parts

"Enrolment and randomization" sub-section at page 6 

The randomisation list had be computer generated - The randomisation list WAS computer generated?

Authors: We have corrected this in the current version of the manuscript. Thank you.

page 13. waiting most of the case their first child - in most cases they are waiting....

Authors: We have corrected this in the current version of the manuscript. Thank you.

---

## [Editor Report · Decision Letter 2]

15 Jul 2021

Couple oriented counselling improves male partner involvement in sexual and reproductive health of a couple: Evidence from the ANRS PRENAHTEST randomized trial.

PONE-D-20-32163R2

Dear Dr. Kengne-Nde,

We’re pleased to inform you that your manuscript has been judged scientifically suitable for publication and will be formally accepted for publication once it meets all outstanding technical requirements.

Kind regards,

Catherine E Oldenburg

Academic Editor

PLOS ONE
---

## [Editor Report · Acceptance letter]

21 Jul 2021

PONE-D-20-32163R2 

Couple oriented counselling improves male partner involvement in sexual and reproductive health of a couple: Evidence from the ANRS PRENAHTEST randomized trial. 

Dear Dr. Kengne-nde:

I'm pleased to inform you that your manuscript has been deemed suitable for publication in PLOS ONE. Congratulations! Your manuscript is now with our production department. 

Kind regards, 

on behalf of

Dr. Catherine E Oldenburg 

Academic Editor

PLOS ONE